# Theranostics Using Indocyanine Green Lactosomes

**DOI:** 10.3390/cancers14153840

**Published:** 2022-08-08

**Authors:** Masaki Kaibori, Kosuke Matsui, Mikio Hayashi

**Affiliations:** 1Department of Surgery, Kansai Medical University, 2-5-1 Shinmachi, Hirakata 573-1191, Japan; 2Department of Physiology, Kansai Medical University, Hirakata 573-1191, Japan

**Keywords:** indocyanine green lactosome, tumor accumulation, photodynamic diagnosis, photodynamic therapy

## Abstract

**Simple Summary:**

Lactosomes™ are biocompatible nanoparticles that can be used for cancer tissue imaging and drug delivery. Lactosomes are amphiphilic micelles in which the particle size can be controlled in the range of 20 to 100 nm. Lactosomes can also be loaded with imaging probes and anticancer agents. Indocyanine green-loaded lactosomes accumulate in cancer tissues and function as a photosensitizer, which simultaneously enables diagnosis and photodynamic therapy. This approach can facilitate the treatment of cancers in unresectable tissues, which can lead to improved quality of life for patients. This review provides an overview of lactosomes with respect to molecular design, accumulation in cancer tissues, and theranostics applications. We also address some outstanding questions and future directions in cancer theranostics.

**Abstract:**

Lactosomes™ are biocompatible nanoparticles that can be used for cancer tissue imaging and drug delivery. Lactosomes are polymeric micelles formed by the self-assembly of biodegradable amphiphilic block copolymers composed of hydrophilic polysarcosine and hydrophobic poly-L-lactic acid chains. The particle size can be controlled in the range of 20 to 100 nm. Lactosomes can also be loaded with hydrophobic imaging probes and photosensitizers, such as indocyanine green. Indocyanine green-loaded lactosomes are stable for long-term circulation in the blood, allowing for accumulation in cancer tissues. Such lactosomes function as a photosensitizer, which simultaneously enables fluorescence diagnosis and photodynamic therapy. This review provides an overview of lactosomes with respect to molecular design, accumulation in cancer tissue, and theranostics applications. The use of lactosomes can facilitate the treatment of cancers in unresectable tissues, such as glioblastoma and head and neck cancers, which can lead to improved quality of life for patients with recurrent and unresectable cancers. We conclude by describing some outstanding questions and future directions for cancer theranostics with respect to clinical applications.

## 1. Introduction

Molecular imaging and theranostic (a fusion of the words “therapy” and “diagnostic”) nanoparticles are effective for the early diagnosis and treatment of cancers. Large-scale industrial, governmental, and academic studies in the field of theranostics have been conducted worldwide. Molecular probes are the cornerstone of imaging and enable early diagnosis and treatment. Molecular probes can be delivered via the enhanced permeability and retention (EPR) effect, which occurs in inflammatory and ischemic diseases as well as in cancers [1]. The EPR effect is a phenomenon in pathological conditions in which nanoparticles with sizes of 30–100 nm accumulate in the interstitium due to vascular leakage [2]. Further, the lymph system surrounding the tumor grows too slowly to exclude nanoparticles from tumor tissues [3]. Lactosomes™ are amphipathic polymers that can be manufactured to exhibit particles of sizes in the range of 20 to 100 nm [4,5,6]. In addition, the shape of polymer micelles can be accurately controlled [7,8]. Lactosomes can be loaded with labeling agents such as indocyanine green (ICG) [9]. ICG has been approved by the Food and Drug Administration for clinical and research use in humans since 1956 [10]. ICG is widely used in ocular angiography and hepatic function assessment. Furthermore, fluorescence imaging using ICG can delineate the area of the tumor and confirm residual tumor after resection, and, subsequently, ICG can be used for photodynamic therapy (PDT). PDT after surgical removal of most of the tumor is used for “tumor bed sterilization”, that is, selective removal of the remaining tumor with minimal damaging of the essential tissue. However, ICG in the blood tends to bind to albumin, which confines most of the bolus to the intravascular space until albumin-mediated hepatic uptake and subsequent excretion into bile [11,12]. Alternatively, ICG-loaded lactosomes remain stable during long-term blood circulation and can accumulate in tumor tissues, where they can act as photosensitizers that generate reactive oxygen species upon application of light of the appropriate wavelength for ICG excitation, about 780 nm [13]. Furthermore, it is possible to label the surface of lactosomes with antibodies to selectively deliver drugs and genes to cancer cells. Thus, lactosomes may be used for both early diagnosis and subsequent treatment of cancers [14]. The aim of this review is to provide an overview of lactosomes with respect to molecular design, accumulation in cancer tissue, and theranostics applications. We will also address some outstanding questions and future directions for theranostic nanoparticles.

## 2. Molecular Design of Indocyanine Green-Loaded Lactosomes

Polymers formed by ICG-poly-L-lactic acid (PLLA) and AB-type polysarcosine PLLA are amphipathic [4,5,6]. PLLA has been used for therapeutic applications, such as in materials for osteosynthesis, as a highly biocompatible and biodegradable hydrophobic polymer. Polysarcosine is a highly hydrophilic polypeptide, with a base material that does not adsorb non-specifically into tissues and cells, and is also biodegradable [15,16]. This amphipathic polymer forms lactosomes in water by self-assembly of PLLA via intermolecular interaction forces. The particle size of the resultant lactosomes can also be controlled. Figure 1 shows the structure of the amphipathic block polymer and aggregates formed by self-assembly [9]. The size and pharmacokinetics of polymer micelles can be accurately controlled by adjusting the numbers of hydrophilic and hydrophobic blocks as well as the amount of polylactic acid. The micelles are formed by linear (AB-type) and trifurcated (A_3_B-type) polymers, with diameters of 35 nm and 22 nm, respectively [7,8]. The half-lives of the AB-type and A_3_B-type polymers in the bloodstream of mice were determined to be 17.2 and 4.3 h, respectively [7]. The hydrodynamic diameters are tunable by varying the mixing ratio of AB-type and A_3_B-type polymers [8]. Additionally, the sizes of micelles composed of AB-type polymers can be adjusted to 100 nm by incorporating PLLA into the hydrophobic core [5]. However, the production of anti-AB-type lactosome IgM significantly increased at 52 nm, implying that the immune system was triggered [7]. Makino et al. carried out chemical modification of the terminal end of PLLA with ICG (ICG-PLLA) [4]. Tsujimoto et al. demonstrated that ICG-loaded lactosomes (ICGm) prepared from ICG-PLLA and AB-type polysarcosine-PLLA, with particle sizes of 40–50 nm, selectively accumulated in tumor tissues [9]. The photoacoustic signal originating from ICGm was increased at 18 h after injection in a mouse model with subcutaneous tumors, indicating efficient accumulation of ICGm in the tumor [13].

### 2.1. Accumulation of Lactosomes in Cancer Tissues

Recent research efforts have focused on the development of molecular probes based on nanoparticles that rely on the EPR effect [2,3]. Nanoparticles containing therapeutic agents such as radioactive and fluorescent agents, as well as magnetic substances, can accumulate in cancer tissues and allow for molecular imaging to sensitively track the effect of treatment. Makino et al. demonstrated the accumulation of ICG-loaded lactosomes in cancer tissue using near-infrared fluorescence imaging in various mouse transplant models of cancer (i.e., subcutaneous, liver, lung, large intestine, and brain) [4,5]. Figure 2 shows the results of fluorescence imaging of orthotopic transplantation and metastatic models of various organs [4,17]. These fluorescence images were obtained at 24 h after intravenous administration of ICG-lactosomes. The fluorescence signal from ICG-lactosomes was detected in various solid tumors in which luceferin-luceferase-induced luminescence-labeled tumor cells grew.

### 2.2. Theranostics Application with ICG-Lactosomes

Biodegradable nanoparticles (ICG-lactosomes) can carry biocompatible cyanine dyes for high biopermeability, near-infrared fluorescence, and photosensitivity, which allows for tumor selectivity and local retention. Combining ICG-lactosomes with a near-infrared light camera allows for: (1) clear delineation of the tumor area during surgery, (2) confirmation of residual tumor after resection to support additional resection if necessary, and (3) a shift to photodynamic therapy to treat only the tumor site.

Although fluorescence imaging has been used to assess the pharmacokinetics of ICG-lactosomes in vivo, this method provides no depth-resolved information. As shown in Figure 3, Tsunoi et al. applied photoacoustic imaging to visualize the depth distribution of ICG-lactosomes in a mouse model of subcutaneous tumor [13]. ICG was illuminated during photoacoustic imaging using a short laser pulse with a wavelength of 796 nm. The laser light was absorbed by ICG, resulting in a rise in temperature and a subsequent initial increase in pressure [18]. The rise in pressure propagated as a photoacoustic wave and was finally detected by a single-element lead zirconate titanate film ultrasound sensor. The depth profile is deduced from the temporal waveform, using the knowledge of the sound velocity in the tissue. The photoacoustic signal originating from the ICG-lactosomes rapidly decreased after PDT, indicating photobleaching by the PDT reaction in the tumor.

The mechanism of the antitumor effects during PDT has previously been reported. First, the photosensitizer, which accumulates in the cancer tissue, is irradiated with a laser light. A photon of the right wavelength is absorbed by the photosensitizer, such as ICG, so that ICG passes from the singlet ground state to an excited singlet state. From there, it passes to the lowest triplet state by intersystem-crossing. The lowest triplet state can then transfer its energy to the surroundings in a collisional process involving molecular oxygen, leading to the generation of reactive oxygen species. The presence of reactive oxygen species may then lead to apoptosis as well as necrosis or other forms of cell death [19]. This system is generally applicable to solid tumors, because it relies on the EPR effect in the neovascularization but does not require a specific antibody for tumor cells. Therefore, the approach can be used to treat cancer cells in unresectable regions, such as parts of the brain in glioblastoma or the tissue near the carotid artery in head and neck cancer. Furthermore, the drug can also be manufactured at a lower cost than antibody-based therapies. This cancer treatment system using near-infrared light with ICG-lactosomes consists of the following elements: (1) a near-infrared fluorescent agent such as ICG-lactosomes, which can accumulate in the cancer and are composed of biodegradable molecules [9]; (2) a near-infrared semiconductor laser irradiator device [9]; and (3) a device that can visualize near-infrared light or a photoacoustic signal during surgery [4,13].

Tsukanishi et al. demonstrated that ICG-lactosomes exhibited effective antitumor effects when combined with laser irradiation [20]. Microscopic examination revealed that MDA-MB-231 cells, a human breast cancer cell line, disappeared after 48 h following PDT [17] (Figure 4). Cell viability was significantly lower in the PDT group compared to the laser-irradiated group.

Funayama et al. also showed that semiconductor laser irradiation was able to treat bone metastases in a mouse model [21] (Figure 5). Near-infrared light can penetrate deep into living tissue. Irradiation by near-infrared light applied from outside the body had antitumor effects on bone metastases. Furthermore, PDT with ICG-lactosomes has also been shown to suppress tumor growth in bone metastatic tumors [17,21].

Tsujimoto et al. reported in vivo imaging using ICG-lactosomes to visualize peritoneal dissemination through the abdominal wall in a mouse model of gastric cancer. While a specific signal was not detected with ICG alone, PDT with ICG-lactosomes showed a significant improvement in survival rate (Figure 6) [9]. The accumulation of ICG-lactosomes was also confirmed in a mouse model of lymph node metastasis with gastric cancer [22] (Figure 7). PDT with ICG-lactosomes was shown to induce apoptosis and inhibit growth of metastatic cancer in lymph nodes. Moreover, our group previously demonstrated that PDT exhibited toxic effects on HuH-7 cells, a human hepatocellular carcinoma cell line, upon treatment with ICG-lactosomes but not ICG alone. Near-infrared fluorescence imaging revealed that ICG-lactosomes accumulated in xenograft tumors and that tumor growth was suppressed by PDT [23]. Furthermore, we confirmed that the fluorescence intensity of the tumor immediately decreased after PDT and that subsequent PDT had greater antitumor effects on a xenograft tumor of gallbladder cancer cells, suggesting that ICG-lactosomes could be useful for theranostics applications [24] (Figure 8).

ICG-lactosomes were also examined for use in a combined tumor therapy system using a combination of photothermal therapy and PDT treatment. Nomura et al. developed a temperature-feedback laser system that maintained the temperature of the irradiated target at a constant level during irradiation [25]. Using this system, the researchers demonstrated complete eradication of the tumor at a surface temperature above 43 °C during photothermal therapy with PDT, regardless of the fluence rate [26]. Furthermore, the authors confirmed that the temperature increase in normal tissue was negligible with photothermal therapy, indicating that ICG-lactosomes tended to accumulate in the tumor (Figure 9).

## 3. Discussion

Treatment of maxillofacial tumors using a photosensitizer-labeled antibody with a similar technology has been approved in the United States [27]. However, such antibody treatments are costly and require antigens that are specific to cancer cells. The intraoperative fluorescence diagnostic technique can accurately identify the excision range. Near-infrared light allows for the diagnosis of deeper residual tumors and lymph node metastases. While ICG has been used to identify sentinel lymph nodes and blood vessels in resected organs, as well as to evaluate blood flow during surgery [28], it is, however, limited by poor tumor accumulation.

ICG-containing nanoparticles that accumulate in the tumor tissue via the EPR effect are excited by near-infrared light, which allows for the combination of photoacoustic diagnosis with photodynamic and photothermal therapy for the treatment of cancer. This system exhibits several features that warrant further discussion. First, ICG-lactosomes allow for visualization of not only the surface layers of cancer tissue but also deeper regions to a depth of a few millimeters [13]. While a depth of a few millimeters is too shallow for clinical applications, the imaging depth can be increased to centimeters by sacrificing spatial resolution [29]. Second, ICG-lactosomes are administered systemically via intravenous injections or direct application to the lesion. The visualization of lesions facilitates accurate operator guidance and fine manipulation, enabling less invasive treatments. Third, ICG-lactosomes function as photosensitizers, which simultaneously enable not only fluorescence diagnosis but also PDT, and this suggests the possibility of applying PDT immediately following a diagnosis. Establishing a highly accurate diagnostic technique for cancer, as presented in this review, enables further resection of marginally located cancers that are not completely resected by surgery, thereby improving prognosis. Clinical studies have confirmed that near-infrared fluorescence imaging with ICG is safe and efficient in pediatric urology using laparoscopy with robotics [30]. The most common applications in pediatric surgery include varicocele repair, difficult cholecystectomy, and partial nephrectomy as well as lymphoma and ovarian tumors [31]. Finally, it has been suggested that the cancer stroma acts as a barrier for drugs that leak from blood vessels. Thus, a therapeutic approach targeting the cancer stroma using the anti-insoluble fibrin antibody may help to overcome the insufficiency of the EPR effect for clinical applications to treat solid cancers [32].

## 4. Conclusions

The current review introduces efforts to develop a theranostic technique for cancer using ICG-lactosome nanoparticles. Lactosomes are amphipathic polymers that can be manufactured with accurate size control. Lactosomes may be loaded with hydrophobic imaging probes and photosensitizers, such as ICG. ICG-loaded lactosomes remain stable during long-term circulation in the blood; accumulate in the tumor tissue via the EPR effect; are excited by near-infrared light; and allow for a combination of photoacoustic diagnosis with photodynamic and photothermal therapy. Clinical studies have confirmed that near-infrared fluorescence imaging with ICG is safe and efficient using laparoscopy with robotics. A therapeutic approach targeting the cancer stroma using the anti-insoluble fibrin antibody may help to overcome the insufficiency of the EPR effect for clinical applications to treat solid cancers.

## Figures and Tables

**Figure 1 cancers-14-03840-f001:**
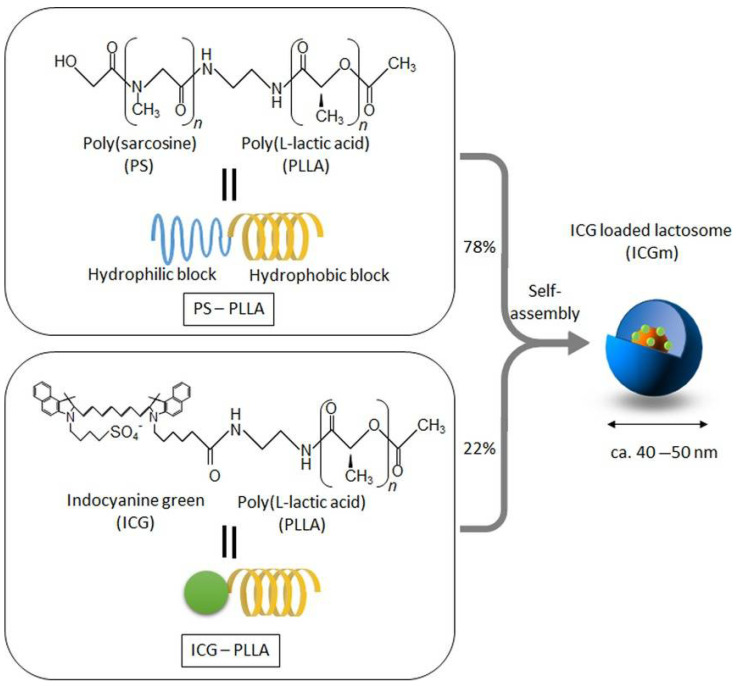
The structure of indocyanine green (ICG)-loaded lactosomes (ICGm’s). ICGm’s are formed by molecular assembly with hydrophobic helical poly-L-lactic acid (PLLA) and hydrophilic polysarcosine (PS) amphiphilic block polydepsipeptide, which include ICG-labeled PLLA in the hydrophobic inner core. (Reprinted from Tsujimoto et al. [9]).

**Figure 2 cancers-14-03840-f002:**
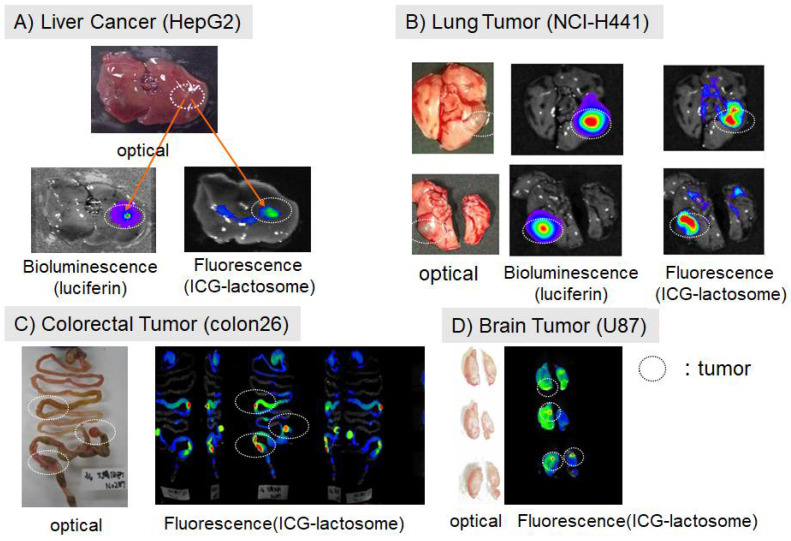
Accumulation of ICG-lactosomes in mouse cancer models. Images were obtained at 24 h after intravenous administration of ICG-lactosomes via the tail vein. The cancer cells carried the luciferase reporter gene. The ICG fluorescence sites overlapped at the luciferin bioluminescence sites, which are a marker of cancer cells. Cancer tissue was confirmed by luciferin bioluminescence and macroscopic findings. (Reproduced from Ozeki and Hara [17]).

**Figure 3 cancers-14-03840-f003:**
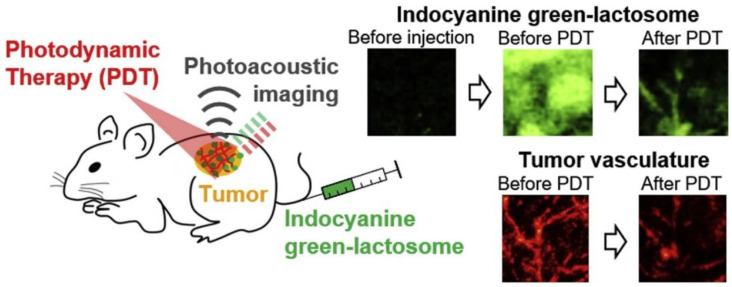
Schematic of photodynamic therapy (PDT) and photoacoustic imaging with ICG-lactosomes. Photoacoustic images (3.9 × 3.9 mm^2^) show the accumulation of ICG-lactosomes in the tumor (before PDT). The photoacoustic probe was automatically scanned over a 3.9 mm × 3.9 mm region of interest with a step size of 50 and 150 μm for imaging of blood vessels and ICG distribution, respectively. The wavelength of the short laser pulse was 532 nm for photoacoustic imaging of oxy- and deoxy-hemoglobin in blood vessels. The laser pulse at 796 nm was used for ICG imaging. Results of the fluorescence imaging of ICG showed that the photoacoustic imaging decreased by only 0.78% in average fluorescence intensity originating from ICG in the tumor (data not shown), indicating a negligible photobleaching by the laser light for photoacoustic imaging. After PDT, the photoacoustic signal was drastically decreased, suggesting photobleaching in the tumor. (Reprinted from Tsunoi et al. [13] with permission from Elsevier).

**Figure 4 cancers-14-03840-f004:**
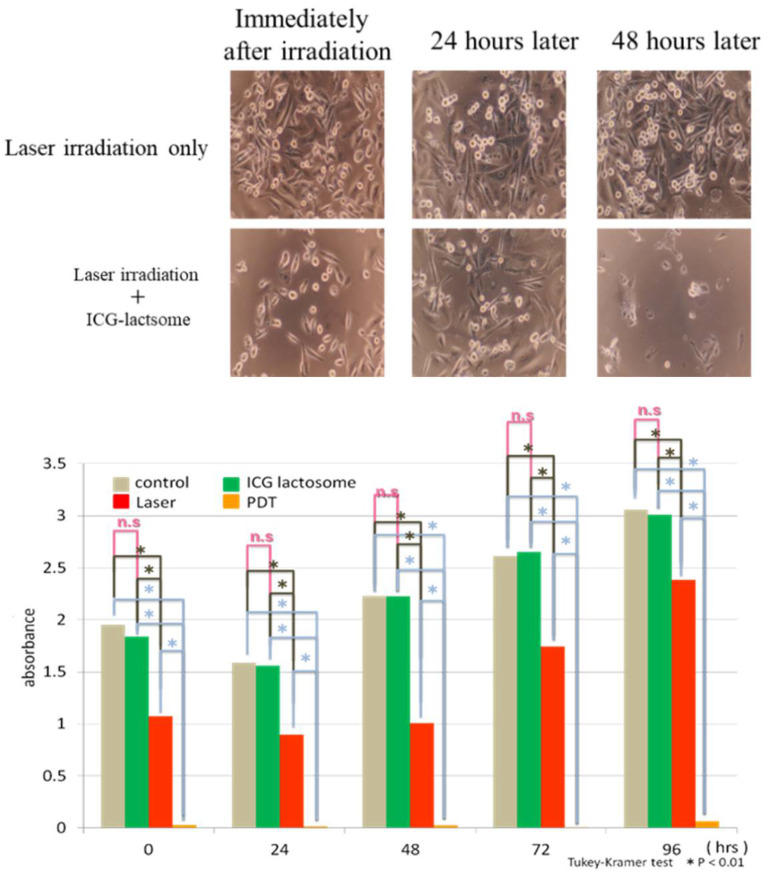
Tumor cell proliferation of the human breast cancer cell line MDA-MB-231 is suppressed by ICG-lactosomes. ICG-lactosomes combined with laser irradiation abolished cells (lower panels). Mean absorbance values indicate cell viability in the WST-1 assay. The laser-irradiated group (*red*) showed significantly lower cell viability than the control and ICG-lactosome alone (*green*) groups. The fluence rate and irradiation period were set to 298 mW/cm^2^ and 60 s, respectively, corresponding to a fluence of 17.9 J/cm^2^. Furthermore, the photodynamic therapy (PDT) group (*yellow*) showed drastically lower cell viability compared to the laser group [20]. n.s, not significant; * *p* < 0.01. (Reproduced from Ozeki and Hara [17]).

**Figure 5 cancers-14-03840-f005:**
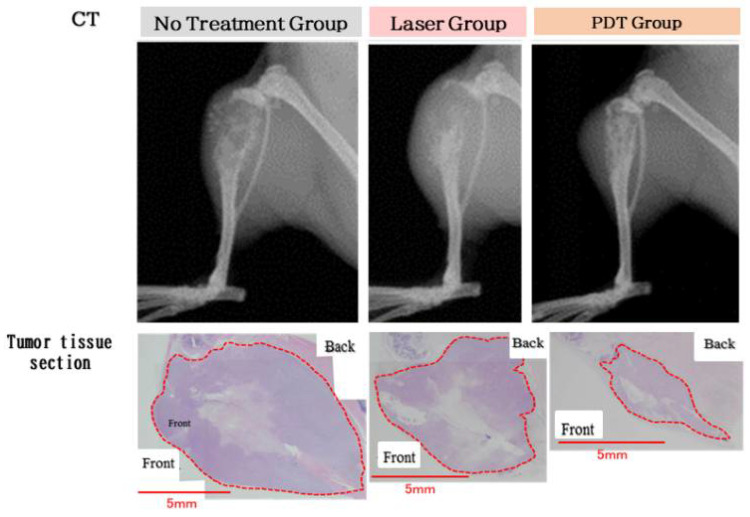
Photodynamic therapy (PDT) in bone metastases. The photodynamic therapy group underwent laser irradiation 24 h after ICG-lactosome administration at 1, 3, and 5 weeks after tumor cell transplantation. The area of osteolysis was compared using computed tomography imaging at 7 weeks after tumor cell transplantation. The area surrounded by the dotted line indicates tumor cells. The PDT group showed smaller tumor areas than the laser group. (Reproduced from Ozeki and Hara [17]).

**Figure 6 cancers-14-03840-f006:**
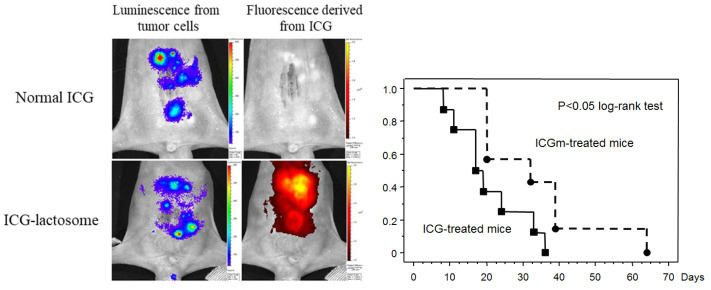
Photodynamic therapy of peritoneal dissemination in gastric cancer. (**Left**) Comparison between ICG and ICG-lactosomes in terms of luminescence and ICG-fluorescence from tumors. (**Right**) Survival rate after photodynamic therapy (PDT) in mice treated with ICG or ICG-lactosomes. PDT with ICG-lactosomes resulted in significantly improved survival rates compared to that with ICG alone. (Reproduced from Tsujimoto et al. [9]).

**Figure 7 cancers-14-03840-f007:**
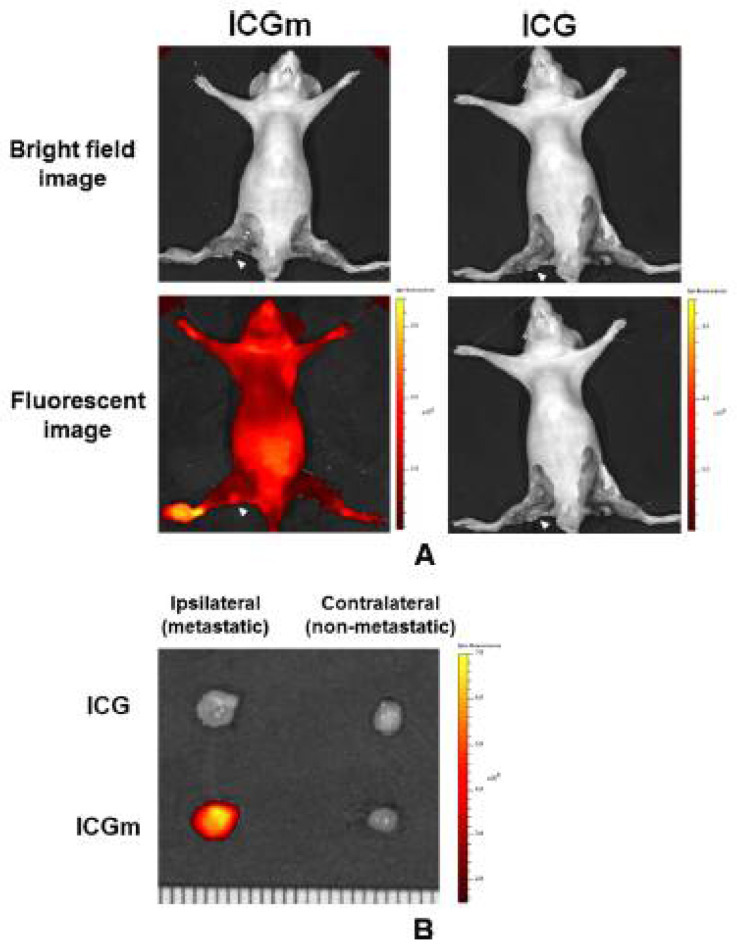
Photodynamic diagnosis of lymph node metastasis. (**A**) In vivo images of the popliteal lymph nodes (*arrowheads*). (**B**) Ex vivo images of the bilateral popliteal lymph nodes. (Reprinted from Tsujimoto et al. [22] with permission from Springer Nature).

**Figure 8 cancers-14-03840-f008:**
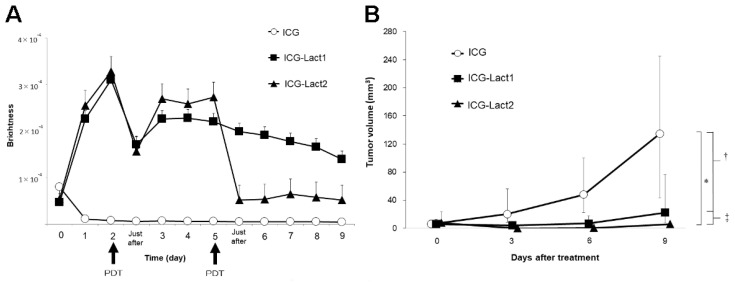
Photodynamic therapy in gallbladder cancer. (**A**) Accumulation of fluorescence imaging in mouse subcutaneous tumors after administration of ICG or ICG-lactosomes was observed daily using in vivo imaging systems. On the second day following administration, PDT was performed in groups with ICG (*n* = 5, open circles) and single (ICG-Lact1, *n* = 5, closed squares) or double PDT with ICG-lactosomes (ICG-Lact2, *n* = 5, closed triangles). The second irradiation was performed on day 5. (**B**) Effect of PDT on tumor growth in mice with subcutaneous tumors treated with ICG and ICG-lactosomes. On the second day following administration, PDT was performed in the ICG (*n* = 8, open circles) and ICG-Lact1 (*n* = 8, closed squares) groups. The second PDT was performed 3 days after the first PDT in the ICG-Lact2 group (*n* = 8, closed triangles). * *p* < 0.001 between the ICG and ICG-Lact2 groups. ^†^ *p* < 0.05 between the ICG and ICG-Lact1 groups. ^‡^
*p* < 0.05 between the ICG-Lact1 and ICG-Lact2 groups. (Reproduced from Hishikawa et al. [24]).

**Figure 9 cancers-14-03840-f009:**
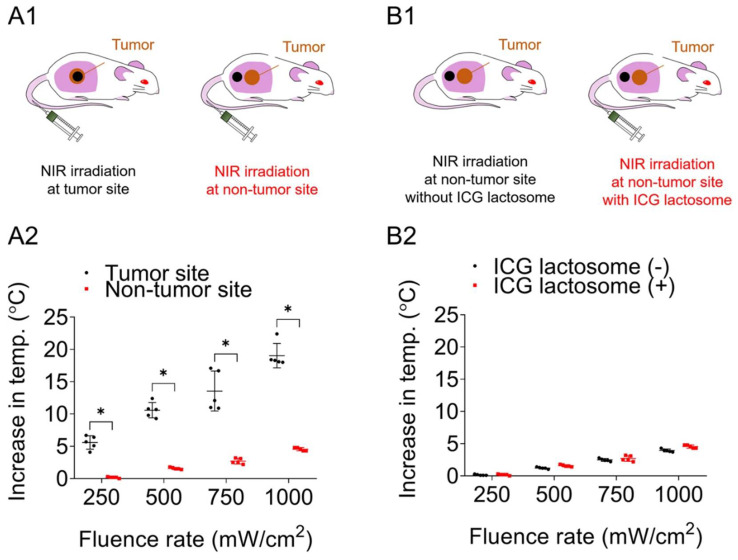
(**A1**) An increase in temperature at the near-infrared (NIR) irradiation spot on the tumor and non-tumor sites in mice administered ICG-lactosomes. The irradiation points are indicated by black circles. (**A2**) The increase in temperature at the tumor site was greater than that at the non-tumor site. * *p* < 0.01. (**B1**) Black circles indicate the NIR irradiation points on the non-tumor site in mice with or without ICG-lactosome administration. (**B2**) There was no difference in temperature increase at the non-tumor site with or without ICG-lactosome administration. (Reprinted from Nomura et al. [26]).

## Data Availability

Not applicable.

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
