# Peer review of "Theranostics Using Indocyanine Green Lactosomes"

_cancers, 2022, doi:10.3390/cancers14153840_

Round 1

Reviewer 1 Report

Dear Authors, you have improved this manuscript with some very well done corrections. Nevertheless, this manuscript ist still not a proper review but a nice summary of articles. Furthermore, you repeat some information(  compare line 48 and lines 68-69; line 91 and line 47). Additionaly, you just copied the text from the articles line 167- our group demonstrated...)

I am sorry, but I can not recommend this version of your manuscript for this journal.

Author Response

We thank Reviewer #1 for providing valuable comments, which helped to improve the quality of our manuscript. We addressed the reviewer’s comments as described in our response below.

Comments from reviewer #1:

Comment 1:

Dear Authors, you have improved this manuscript with some very well done corrections. Nevertheless, this manuscript ist still not a proper review but a nice summary of articles. Furthermore, you repeat some information(  compare line 48 and lines 68-69; line 91 and line 47). Additionaly, you just copied the text from the articles line 167- our group demonstrated...)

Response 1:

Per the reviewer’s comment, we deleted the repeat information and revised the corresponding text as follows (page 7, lines 188): “PDT with ICG-lactosomes was shown to induce apoptosis and inhibit growth of meta-static cancer in lymph nodes. Moreover, our group previously demonstrated that PDT exhibited toxic effects on HuH-7 cells, a human hepatocellular carcinoma cell line, upon treatment with ICG-lactosomes, but not ICG alone. Near-infrared fluorescence imaging revealed that ICG-lactosomes accumulated in xenograft tumors, and that tu-mor growth was suppressed by PDT [20]. Furthermore, we confirmed that the fluores-cence intensity of the tumor immediately decreased after PDT and that subsequent PDT had greater antitumor effects on a xenograft tumor of gallbladder cancer cells, suggesting that ICG-lactosomes can be useful for theranostics applications [21]”

Again, we thank Reviewer #1 for providing helpful comments, which has allowed us to greatly improve the quality of our manuscript.

Reviewer 2 Report

The study, titled “Theranostics using indocyanine green lactosomes.” This review discusses the promising results of using Lactosome based nanoparticles that are biocompatible that can be used for cancer tissue imaging and drug delivery. Overall, I would recommend the publication in Cancers if the authors can kindly address my comment and the following minor comments:

1.     Line 24: Repeating the sentences in many places, “particle size can be controlled in the range of 20 to 200 24 nm with an accuracy of 5 nm”

2.  Line 140: Repeating the same sentences in many places is “clarified.” As suggested, use an alternative word.

3.    The explanations in Figures 4, 5, and 6 were limited as suggested to emphasize the explanation.

4.     In the discussion section, as suggested, compare the material merits and demerits of ICG-lactosomes using a table format.

5.  Outstanding questions and future directions in cancer theranostics for ICG-lactosomes Discuss in the conclusion section, as recommended.

Author Response

We thank Reviewer #2 for providing valuable comments, which helped improve the quality of the revised manuscript. We addressed the reviewer’s comments as described in our responses below.

Comments from reviewer #2:

Comment 1:

Line 24: Repeating the sentences in many places, particle size can be controlled in the range of 20 to 200 24 nm with an accuracy of 5 nm”

Response 1:

Per the reviewer’s suggestion, we deleted the following repeat information (page 10, lines 270): "Lactosomes are amphipathic polymers that can be manufactured with accurate size control.”

Comment 2:

Line 140: Repeating the same sentences in many places is clarified.” As suggested, use an alternative word.

Response 2:

Per the reviewer’s suggestion, we revised the corresponding text as follows:

(page 3, line 96) “Makino et al. demonstrated the accumulation of ICG-loaded lactosomes in cancer tissue using near-infrared fluorescence imaging in various mouse transplant models of cancer ”

(page 6, lines 171) “Funayama et al. also showed that semiconductor laser irradiation was able to treat bone metastases in a mouse model”

Comment 3:

The explanations in Figures 4, 5, and 6 were limited as suggested to emphasize the explanation.

Response 3:

Per the reviewer’s comment, we added the following sentences to the revised manuscript:

(page 5, lines 158): Microscopic examination revealed that MDA-MB-231 cells, a human breast cancer cell line, disappeared after 48 hours following PDT [14] (Figure 4). Cell viability was sig-nificantly lower in the PDT group compared to the laser-irradiated group. ”

(page 6, lines 174): “Furthermore, PDT with ICG-lactosomes has also been shown to suppress tumor growth in bone metastatic tumors”

(page 7, lines 183): “Tsujimoto et al. reported in vivo imaging using ICG-lactosomes to visualize peritoneal dissemination through the abdominal wall in a mouse model of gastric cancer. While a specific signal was not detected with ICG alone, PDT with ICG-lactosomes showed a significant improvement in survival rate”

Comment 4:

In the discussion section, as suggested, compare the material merits and demerits of ICG-lactosomes using a table format.

Response 4:

Per the reviewer’s comment, we included a discussion of the merits and demerits of ICG-lactosomes as follows:

(page 9, lines 245): “ICG-containing nanoparticles that accumulate in the tumor tissue via the EPR effect are excited by near-infrared light, which allows for the combination of photoacoustic diagnosis with photodynamic and photothermal therapy for the treatment of cancer.”

(page 10, lines 263): “Finally, it has been suggested that the cancer stroma acts as a barrier for drugs that leak from blood vessels. Thus, a therapeutic approach targeting the cancer stroma using the anti-insoluble fibrin antibody may help overcome the insufficiency of the EPR effect for clinical applications to treat solid cancers [29].”

Comment 5:

Outstanding questions and future directions in cancer theranostics for ICG-lactosomes Discuss in the conclusion section, as recommended.

Response 5:

Per the reviewer’s comment, we added the following text to the Conclusions (page 10, lines 272): “ICG-loaded lactosomes remain stable during long-term circulation in the blood, accumulate in the tumor tissue via the EPR effect, are excited by near-infrared light, and allow for a combination of photoacoustic diagnosis with photodynamic and photo-thermal therapy.”

(page 10, lines 277): “A therapeutic approach targeting the cancer stroma using the anti-insoluble fibrin antibody may help overcome the insufficiency of the EPR effect for clinical applications to treat solid cancers.”

Again, we thank Reviewer #2 for providing helpful comments, which have allowed us to greatly improve the quality of our manuscript.

Reviewer 3 Report

Authors addressed my comments. The manuscript is suitable for pubblication.

Author Response

We thank Reviewer #3 for providing valuable comments, which helped improve the quality of the revised manuscript.

This manuscript is a resubmission of an earlier submission. The following is a list of the peer review reports and author responses from that submission.

Round 1

Reviewer 1 Report

Dear Authors,

this manuscript is not a review but a nice summary of 18 articles. 

Have you already received publisher´s permission to use all the images?

I am sorry, but I will not recommend this version of your manuscript for this journal.

Author Response

We want to thank Reviewer #1 for providing valuable comments, which helped improve the quality of the manuscript.  As shown in the responses, all comments were taken into consideration.

Comments from reviewer #1:

Comment 1:

This manuscript is not a review but a nice summary of 18 articles. Have you already received publisher´s permission to use all the images? I am sorry, but I will not recommend this version of your manuscript for this journal.

Response 1:

In accordance with the suggestion, we received the publisher’s permission to use all the images. The following sentences were also added in the Figure legends:

Page 3, line 87, “(Reproduced from Tsujimoto et al. [7].)”

Page 6, line 175, “(Reproduced from Tsujimoto et al. [7].)”

Page 7, lines 178, “(Reproduced from Tsujimoto et al. [14] with permission from Springer Nature.)”

Page 7, line 192, “(Reproduced from Hishikawa et al. [16].)”

Page 8, lines 207, “(Reproduced from Nomura et al. [17].)”

We want to thank you for providing positive and helpful comments, and we hope that the manuscript is now acceptable for publication.

Reviewer 2 Report

The study, titled “Theranostics using indocyanine green lactosomes.” This review discusses the promising results of using Lactosome based nanoparticles that are biocompatible that can be used for cancer tissue imaging and drug delivery. The paper in its current form is unsuitable for publication; significant changes necessitate rewriting the paper. This review paper should be updated by answering the following questions. –

Major comments -

Abstract-

  1. The significance of Lactosome-based nanoparticle shape and particle size is unclear, so the abstract section has been revised.

Introduction –

  1. The English language must be completely rewritten by a native English speaker or equivalent who is also familiar with photobiology.
  2. ICG-lactosomes, which are Indocyanine Green (ICG) derivatives, have been published. As a result, it is suggested that the changes made and how to overcome the limitations of the ICG-lactosomes be highlighted more in this section.
  3. The introduction section is unclear, as suggested by the revised topic, such as molecular imaging technology, how to overcome the limitation of ICG-lactosomes, the importance of ICG dye, and the importance of lactosomes.
  4. Discuss the size, shape, and impact of ICG-lactosomes nanoparticles on long-term blood circulation.
  5. The importance of PDT reactions and ROS reactions in cancer is added in the introduction section.
  6. The authors discuss the uniqueness of ICG-lactosomes particles.

Materials and methods –

  1. The importance of ICG-lactosome size, aggregation, and stability should be discussed in this section of "Morphological Control Through Molecular Design and Self-Organization."

Results-

  1. The tumor accumulation section is unclear; the authors should explain the important PDT mechanisms for tumor treatment included in this section more clearly.
  2. The literature review discussion is limited, therefore specify clearly.
  3. The figures (1-9) explanation is very poor, as suggested that added recently published papers in the result sections.
  4. In the subtitle of the result part section, the authors should be revised and add a more specified title.
  5. The ICG-lactosomes nanoparticles, according to the authors explanation, are very old. The results of the most recent paper were added to the revised entire result section, as suggested.

Discussion and conclusion.

  1. The explanations for the discussion and conclusion sections are incomplete. As a result, the authors should emphasize the point clearly.

References-

  1. The authors should update all of the references; I couldn't find any recently published papers.
  2. The many sentences missing the references revised the addition of the references section.

Author Response

We want to thank reviewer #2 for providing valuable comments, which helped improve the quality of the revised manuscript. As shown in the responses, all comments were considered.

Comments from reviewer #2:

Comment 1:

Abstract 1: The significance of Lactosome-based nanoparticle shape and particle size is unclear, so the abstract section has been revised.

Response 1:

Based on the suggestion, the following sentences were added in the Abstract (page 1, line 22): “This nanoparticle is a polymeric micelle formed by the self-assembly of biodegradable amphiphilic block copolymers, which are composed of hydrophilic poly-sarcosine, and hydrophobic poly-L-lactic acid chains. The particle size can be controlled in the range of 20 to 200 nm with an accuracy of 5 nm. They may contain hydrophobic labeling and anticancer agents such as indocyanine green.”

Comment 2:

Introduction 1: The English language must be completely rewritten by a native English speaker or equivalent who is also familiar with photobiology.

Response 2:

The manuscript has been thoroughly edited by a native English speaker with expertise in medical imaging technologies.

Comment 3:

Introduction 2: ICG-lactosomes, which are Indocyanine Green (ICG) derivatives, have been published. As a result, it is suggested that the changes made and how to overcome the limitations of the ICG-lactosomes be highlighted more in this section.

Response 3:

Based on your suggestion, the following sentences were added to the Introduction (page 2, line 48): “In addition, the shape of the polymer micelle can be accurately controlled [5,6]. Lactosomes™ may contain hydrophobic labeling and anticancer agents such as indocyanine green (ICG) [7].”

Comment 4:

Introduction 3: The introduction section is unclear, as suggested by the revised topic, such as molecular imaging technology, how to overcome the limitation of ICG-lactosomes, the importance of ICG dye, and the importance of lactosomes.

Response 4:

We added the sentences to describe indocyanine green as a labeling and anticancer agent (page 2, line 49): “Lactosomes™ may contain hydrophobic labeling and anticancer agents such as indocyanine green (ICG) [7]. Using fluorescence imaging with ICG, we can delineate the area of the tumor, confirm residual tumor after resection, and shift to photodynamic therapy (PDT).”

Comment 5:

Introduction 4: Discuss the size, shape, and impact of ICG-lactosomes nanoparticles on long-term blood circulation.

Response 5:

We added the sentences in the Introduction section (pages 1-2, line 43): “To deliver the molecular probes, we utilize the enhanced permeability and retention (EPR) effect, which occurs in inflammatory and ischemic diseases, as well as in the initial growth of cancer [1]. The EPR effect is a phenomenon in pathological conditions, in which nanoparticles of 20–200 nm accumulate in the interstitium due to capillary leakage. Lactosomes™ are amphipathic polymers, which can be manufactured from 20 to 200 nm with an accuracy of 5 nm [2–4].”

Comment 6:

Introduction 5: The importance of PDT reactions and ROS reactions in cancer is added in the introduction section.

Response 6:

Based on your suggestion, the following sentence was added in the Introduction section (page 2, line 52): “ICG-loaded lactosomes™ are stable over long-term blood circulation and enable the accumulation of the photosensitizer that generates reactive oxygen species after the application of laser light to the tumor tissue [8].”

Comment 7:

Introduction 6: The authors discuss the uniqueness of ICG-lactosomes particles.

Response 7:

Based on your suggestion, the following sentence was added in the Introduction section (page 2, line 51): “Using fluorescence imaging with ICG, we can delineate the area of the tumor, confirm residual tumor after resection, and shift to photodynamic therapy (PDT).”

Comment 8:

Materials and methods 1: The importance of ICG-lactosome size, aggregation, and stability should be discussed in this section of "Morphological Control Through Molecular Design and Self-Organization.

Response 8:

We changed the title of the section.

Page 2, line 61, “2. Molecular Design of Indocyanine Green-loaded Lactosomes™”.”

Based on your suggestion, we revised the sentences in this section (page 2, lines 66): “This amphipathic polymer forms lactosomes™ in water by self-assembly of the PLLA via intermolecular interaction forces, and the particle size can be controlled within a range of 20 to 200 nm with an accuracy of 5 nm. Figure 1 indicates the structure of the amphipathic block polymer and the aggregates formed by self-assembly [5–7]. The shape and pharmacokinetics of the polymer micelle can be accurately controlled by compounding the amount of the hydrophilic and hydrophobic blocks and polylactic acid. The micelle is formed by linear (AB-type) and trifurcated (A3B-type) polymers whose diameters are 35 and 20 nm, respectively [5,6]. The A3B-type polymers are considered to be smaller than the AB-type because of the large hydrophilicity. Interestingly, the particle size can continuously vary within the range of 20–35 nm depending on the mixing ratio of the polymers. Additionally, the hydrophobic core can be adjusted up to 200 nm by combining the PLLA with the AB-type polymer. Thus, lactosomes™ can encapsulate hydrophobic agents within. Tsujimoto et al. successfully synthesized ICG-loaded lactosomes™ (ICGm) [7]. The photoacoustic signal originating from the ICGm was increased at 18 hours after injection in a mouse model with subcutaneous tumors, indicating the efficient accumulation of the ICGm in the tumor [8].”

Comment 9:

Results 1: The tumor accumulation section is unclear; the authors should explain the important PDT mechanisms for tumor treatment included in this section more clearly.

Response 9:

Based on your suggestion, the following sentence was added to the Accumulation of Lactosomes™ in Cancer Tissues section (page 3, line 92): “Nanoparticles containing therapeutic agents such as radioactive and fluorescent agents, as well as magnetic substances can accumulate in cancer tissue for molecular imaging, which can sensitively track the effect of treatment.”

We also discussed important PDT mechanisms for tumor treatment on page 5, line 125, “The mechanism for the antitumor effects during photodynamic therapy has been explained. First, the photosensitizer, which accumulates in cancer tissue is irradiated with laser light. The activated photosensitizer reacts with endogenous oxygen, thereby producing singlet oxygen, which causes apoptosis of the tumor cells. The reaction also generates heat, which contributes to the tumor-suppressing effect. This system is generally applicable to solid tumors, because it utilizes the EPR effect in the neovascularization but does not require a specific antibody for the tumor cells.”

Comment 10:

Results 2: The literature review discussion is limited, therefore specify clearly.

Response 10:

We specified the literature (page 4, line 116): “As shown in Figure 3, Tsunoi et al. applied photoacoustic imaging to visualize the depth distribution of ICG-lactosomes™ in a mouse model of subcutaneous tumor [8].”

Comment 11:

Results 3: The figures (1-9) explanation is very poor, as suggested that added recently published papers in the result sections.

Response 11:

Based on your suggestion, the following sentences were added.

On page 3, line 100, “The fluorescence signal from the ICG-lactosomes™ was detected in various solid tumors where the luminescence-labeled tumor cells grew.”

On page 4, line 116, “As shown in Figure 3, Tsunoi et al. applied photoacoustic imaging to visualize the depth distribution of ICG-lactosomes™ in a mouse model of subcutaneous tumor [8]. The photoacoustic signal originating from the ICG-lactosomes™ was rapidly decreased after PDT, indicating photobleaching by the PDT reaction in the tumor.”

On page 5, line 146, “ICG-lactosome™ combined with laser irradiation abolished cells (lower panels). Mean absorbance values indicate cell viability in the WST-1 assay. The laser group (red) showed significantly lower cell viability than the control and ICG lactosome™ alone (green) groups. The photodynamic therapy (PDT) group (yellow) showed significantly lower cell viability compared to the laser group [12].”

On page 6, line 163, “Tsujimoto et al. reported in vivo imaging using ICG-lactosomes™ to visualize peritoneal dissemination through the abdominal wall in a model of gastric cancer. Subsequently, PDT with ICG-lactosomes™ significantly improved the survival rate (Figure 6) [7].”

On page 7, line 177, “Photodynamic diagnosis of lymph node metastasis. (A) In vivo images of the popliteal lymph nodes (arrowheads). (B) Ex vivo images of the bilateral popliteal lymph nodes.”

On page 8, line 203, “The irradiation points are indicated by black circles. (A2) The increase in temperature at the tumor site was greater than that at the non-tumor site. (B1) Black circles indicate the near-infrared irradiation points on the non-tumor site in mice with or without ICG-lactosomes™ administration. (B2) The increase in temperature at the non-tumor site was not different with or without ICG-lactosomes™ administration.”

Comment 12:

Results 4: In the subtitle of the result part section, the authors should be revised and add a more specified title.

Response 12:

Based on your suggestion, the subtitles were revised in the result section.

Page 3, line 88: “3.1. Accumulation of Lactosomes™ in Cancer Tissues”

Page 4, line 108: “3.2. Theranostics Application with ICG-lactosomes™”

Comment 13:

Results 5: The ICG-lactosomes nanoparticles, according to the authors explanation, are very old. The results of the most recent paper were added to the revised entire result section, as suggested.

Response 13:

Based on your suggestion, we added the sentences on page 9, line 231, “Clinical studies have confirmed that near-infrared fluorescence imaging with ICG is safe and efficient in pediatric urology using laparoscopy with robotics [19]. The most common applications in pediatric surgery include varicocele repair, difficult cholecystectomy, and partial nephrectomy, as well as lymphoma and ovarian tumors [20]. In addition to the photosensitizer, 89Zn radiolabeling, antibody variants, cell penetrating peptide, and siRNA all have challenges which must be overcome for successful future clinical development and implementation in cancer theranostics [21].”

The following references were also added:

  1. Esposito, C.; Coppola, V.; Del Conte, F.; Cerulo, M.; Esposito, G.; Farina, A.; Crocetto, F.; Castagnetti, M.; Settimi, A.; Escolino, M. Near-Infrared fluorescence imaging using indocyanine green (ICG): Emerging applications in pediatric urology. J. Pediatr. Urol. 2020, 16, 700–707; DOI:10.1016/j.jpurol.2020.07.008.
  2. Esposito, C.; Settimi, A.; Del Conte, F.; Cerulo, M.; Coppola, V.; Farina, A.; Crocetto, F.; Ricciardi, E.; Esposito, G.; Escolino, M. Image-Guided Pediatric Surgery Using Indocyanine Green (ICG) Fluorescence in Laparoscopic and Robotic Surgery. Front Pediatr. 2020, 8, 314; DOI:10.3389/fped.2020.00314.
  3. Lim, M.S.H.; Ohtsuki, T.; Takenaka, F.; Kobayashi, K.; Akehi, M.; Uji, H.; Kobuchi, H.; Sasaki, T.; Ozeki, E.; Matsuura, E. A Novel 89Zr-labeled DDS Device Utilizing Human IgG Variant (scFv): "Lactosome" Nanoparticle-Based Theranostics for PET Imaging and Targeted Therapy. Life (Basel). 2021, 11, 158; DOI:10.3390/life11020158.

Comment 14:

Discussion and conclusion 1: The explanations for the discussion and conclusion sections are incomplete. As a result, the authors should emphasize the point clearly.

Response 14:

Based on your suggestion, the following sentences were added in the Conclusions section (page 9, line 240): “The current review introduces efforts to develop a theranostic technique for cancer using ICG lactosome™ nanoparticles. Lactosomes™ are amphipathic polymers, which can be manufactured from 20 to 200 nm with an accuracy of 5 nm with accurate shape control. They may contain hydrophobic labeling and anticancer agents such as ICG. The ICG-loaded lactosomes™ are stable over long-term blood circulation and enable accumulation as photosensitizers, which generate reactive oxygen species by photodynamic stimulation in tumors. Clinical studies have confirmed that near-infrared fluorescence imaging with ICG is safe and efficient using laparoscopy with robotics. Furthermore, various modalities with lactosomes will be expected for future clinical development in cancer theranostics.”

Comment 15:

References 1: The authors should update all of the references; I couldn't find any recently published papers.

Response 15:

The following references were added:

  1. Esposito, C.; Coppola, V.; Del Conte, F.; Cerulo, M.; Esposito, G.; Farina, A.; Crocetto, F.; Castagnetti, M.; Settimi, A.; Escolino, M. Near-Infrared fluorescence imaging using indocyanine green (ICG): Emerging applications in pediatric urology. J. Pediatr. Urol. 2020, 16, 700–707; DOI:10.1016/j.jpurol.2020.07.008.
  2. Esposito, C.; Settimi, A.; Del Conte, F.; Cerulo, M.; Coppola, V.; Farina, A.; Crocetto, F.; Ricciardi, E.; Esposito, G.; Escolino, M. Image-Guided Pediatric Surgery Using Indocyanine Green (ICG) Fluorescence in Laparoscopic and Robotic Surgery. Front Pediatr. 2020, 8, 314; DOI:10.3389/fped.2020.00314.
  3. Lim, M.S.H.; Ohtsuki, T.; Takenaka, F.; Kobayashi, K.; Akehi, M.; Uji, H.; Kobuchi, H.; Sasaki, T.; Ozeki, E.; Matsuura, E. A Novel 89Zr-labeled DDS Device Utilizing Human IgG Variant (scFv): "Lactosome" Nanoparticle-Based Theranostics for PET Imaging and Targeted Therapy. Life (Basel). 2021, 11, 158; DOI:10.3390/life11020158.

Comment 16:

References 2: The many sentences missing the references revised the addition of the references section.

Response 16:

We have thoroughly revised the References section.

We want to thank you for providing positive and helpful comments, and we hope that the manuscript is now acceptable for publication.

Reviewer 3 Report

    My suggestions:
    - avoid the use of abbreviation in simple summary
    - check line 16
    - lines 21-23 are difficult to understand
    - In my opinion, the use of "materials and methods" and "results" paragraphs are not pertinent.
    -at the end of introduction section, include the aim of the review
    Could be the method proposed affected by false positive results? Are there other factors inducing neovascularization?
    - check line 86
    What is the resolution limit of this method?
    Lines 97-100 are not intellegible.
    In figure 2 authors should include a negative control (i.e. a normal organ specimen)
    Explain the type of pathological analysis indicated in legend of figure 2
    -check lines 131-134-142-173
    - sentence at lines 141-143 should be supported by experimental evidence
    -PDT therapy and photoacoustic imaging should be explainede in a exaustive manner following a logic order. 
    - In figure 4, it should be included ICG alone
    - In figure legends indicate the reference of the manuscript reporting the figures showed.
    -sentence at line 211 is truncated
    The methods proposed seems very interesting. However,  it should be emphasized the clinical application for several human diseases. The following are two recently published studies showing this aspect:
  • DOI: 10.3389/fped.2020.00314
  • DOI: 10.1016/j.jpurol.2020.07.008In addition, it could be highly appreciated show the evidence acquired in several cancer types.

Author Response

We want to thank reviewer #3 for the valuable comments that helped improve the quality of the revised manuscript. As shown in the responses, all comments were considered.

Comments from reviewer #3:

Comment 1:

avoid the use of abbreviation in simple summary

Response 1:

The simple summary has been thoroughly rewritten to avoid the use of abbreviations.

Comment 2:

check line 16

Response 2:

Based on your suggestion, we revised the sentence on page 1, line 22, “This nanoparticle is a polymeric micelle formed by the self-assembly of biodegradable amphiphilic block copolymers, which are composed of hydrophilic poly-sarcosine, and hydrophobic poly-L-lactic acid chains.”

Comment 3:

lines 21-23 are difficult to understand

Response 3:

We deleted the sentence.

Comment 4:

In my opinion, the use of "materials and methods" and "results" paragraphs are not pertinent.

Response 4:

Based on your suggestion, we changed the titles.

Page 2, line 61, “2. Molecular Design of Indocyanine Green-loaded Lactosomes™”.

Page 3, line 88, “3.1. Accumulation of Lactosomes™ in Cancer Tissues”.

Page 4, line 108, “3.2. Theranostics Application with ICG-lactosomes™”.

Comment 5:

at the end of introduction section, include the aim of the review

Response 5:

Based on your suggestion, we added the sentences on page 2, line 57, “The aim of this review is to provide an overview of lactosomes™ with respect to molecular design, accumulation in cancer tissue, and theranostics applications. We will also address some outstanding questions and future directions for theranostic nanoparticles.”

Comment 6:

Could be the method proposed affected by false positive results? Are there other factors inducing neovascularization?

Response 6:

We explained the enhanced permeability and retention effect in the Introduction section (pages 1-2, line 43): “To deliver the molecular probes, we utilize the enhanced permeability and retention (EPR) effect, which occurs in inflammatory and ischemic diseases, as well as in the initial growth of cancer [1]. The EPR effect is a phenomenon in pathological conditions, in which nanoparticles of 20–200 nm accumulate in the interstitium due to capillary leakage. Lactosomes™ are amphipathic polymers, which can be manufactured from 20 to 200 nm with an accuracy of 5 nm [2–4].”

Comment 7:

check line 86

Response 7:

We revised the sentence on page 3, line 89, “Recent research has focused on…”

Comment 8:

What is the resolution limit of this method?

Response 8:

Based on your suggestion, we revised the sentence on page 4, line 123, “Photoacoustic images (3.9 x 3.9 mm2) show the accumulation of the ICG-lactosomes™ in the tumor (Before PDT).”

Comment 9:

Lines 97-100 are not intellegible.

Response 9:

Based on your suggestion, we revised the sentence on page 3, line 94, “Makino et al. clarified the accumulation of the ICG-loaded lactosomes™ in cancer tissue using near-infrared fluorescence imaging in various mouse transplant models of cancer (i.e., subcutaneous, liver, lung, large intestine, and brain). [2,3].”

Comment 10:

In figure 2 authors should include a negative control (i.e. a normal organ specimen)

Response 10:

Based on your suggestion, we added the sentence on page 4, line 105, “The cancer cells carried the luciferase reporter gene.”

Comment 11:

Explain the type of pathological analysis indicated in legend of figure 2

Response 11:

Based on your suggestion, we revised the sentence on page 4, line 106, “Cancer tissue was confirmed via the luminescence method (luciferin) and macroscopic findings.”

Comment 12:

check lines 131-134-142-173

Response 12:

Based on your suggestion, we revised the following sentences.

Page 4, line 118, “The photoacoustic signal originating from the ICG-lactosomes™ was rapidly decreased after PDT, indicating photobleaching by the PDT reaction in the tumor.”

Page 5, line 125, “The mechanism for the antitumor effects during photodynamic therapy has been explained .”

Page 5, line 131, “Therefore, the approach enables the treatment of metastatic lesions near the margin, which cannot be completely resected, and thus, leads to improved quality of life in patients with recurrent and unresectable disease.”

Page 6, line 163, “Tsujimoto et al. reported in vivo imaging using ICG-lactosomes™ to visualize peritoneal dissemination through the abdominal wall in a model of gastric cancer. Subsequently, PDT with ICG-lactosomes™ significantly improved the survival rate (Figure 6) [7].”

Comment 13:

sentence at lines 141-143 should be supported by experimental evidence

Response 13:

Based on your suggestion, we revised the sentence on page 5, line 131, “Therefore, the approach enables the treatment of metastatic lesions near the margin, which cannot be completely resected, and thus, leads to improved quality of life in patients with recurrent and unresectable disease.”

Comment 14:

PDT therapy and photoacoustic imaging should be explainede in a exaustive manner following a logic order.

Response 14:

We revised the sentence on page 4, line 122, “Schematic figure of photodynamic therapy (PDT) and photoacoustic imaging with ICG-lactosomes™.”

On page 5, line 138, (2) a near-infrared semiconductor laser irradiator device, and (3) a device that can visualize near-infrared light during surgery.”

Comment 15:

In figure 4, it should be included ICG alone

Response 15:

Based on your suggestion, we revised the sentence on page 5, line 147, “The laser group (red) showed significantly lower cell viability than the control and ICG lactosome™ alone (green) groups.”

Comment 16:

In figure legends indicate the reference of the manuscript reporting the figures showed.

Response 16:

In accordance with your suggestion, we indicated the reference numbers in figure legends:

Page 3, line 87, “(Reproduced from Tsujimoto et al. [7].)”

Page 4, line 124, “After PDT, the photoacoustic signal was drastically decreased [8].”

Page 5, line 148, “The photodynamic therapy (PDT) group (yellow) showed significantly lower cell viability compared to the laser group [12].”

Page 6, line 175, “(Reproduced from Tsujimoto et al. [7].)”

Page 7, lines 178, “(Reproduced from Tsujimoto et al. [14] with permission from Springer Nature.)”

Page 7, line 192, “(Reproduced from Hishikawa et al. [16].)”

Page 8, lines 207, “(Reproduced from Nomura et al. [17].)”

Comment 17:

sentence at line 211 is truncated

Response 17:

We revised the sentence on pages 7-8, line 193, “Nomura et al. examined the therapeutic effects of PDT with ICG-lactosomes™ while changing the fluence rate and irradiation time in various combinations and found that maintaining the tumor temperature above 43  °C during near-infrared irradiation was a more reliable determinant of therapeutic effect.”

Comment 18:

The methods proposed seems very interesting. However,  it should be emphasized the clinical application for several human diseases. The following are two recently published studies showing this aspect:

  • DOI: 10.3389/fped.2020.00314
  • DOI: 10.1016/j.jpurol.2020.07.008

In addition, it could be highly appreciated show the evidence acquired in several cancer types.

Response 18:

Based on your suggestion, we added the sentences on page 9, line 231, “Clinical studies have confirmed that near-infrared fluorescence imaging with ICG is safe and efficient in pediatric urology using laparoscopy with robotics [19]. The most common applications in pediatric surgery include varicocele repair, difficult cholecystectomy, and partial nephrectomy, as well as lymphoma and ovarian tumors [20]. In addition to the photosensitizer, 89Zn radiolabeling, antibody variants, cell penetrating peptide, and siRNA all have challenges which must be overcome for successful future clinical development and implementation in cancer theranostics [21].”

The following references were also added:

  1. Esposito, C.; Coppola, V.; Del Conte, F.; Cerulo, M.; Esposito, G.; Farina, A.; Crocetto, F.; Castagnetti, M.; Settimi, A.; Escolino, M. Near-Infrared fluorescence imaging using indocyanine green (ICG): Emerging applications in pediatric urology. J. Pediatr. Urol. 2020, 16, 700–707; DOI:10.1016/j.jpurol.2020.07.008.
  2. Esposito, C.; Settimi, A.; Del Conte, F.; Cerulo, M.; Coppola, V.; Farina, A.; Crocetto, F.; Ricciardi, E.; Esposito, G.; Escolino, M. Image-Guided Pediatric Surgery Using Indocyanine Green (ICG) Fluorescence in Laparoscopic and Robotic Surgery. Front Pediatr. 2020, 8, 314; DOI:10.3389/fped.2020.00314.
  3. Lim, M.S.H.; Ohtsuki, T.; Takenaka, F.; Kobayashi, K.; Akehi, M.; Uji, H.; Kobuchi, H.; Sasaki, T.; Ozeki, E.; Matsuura, E. A Novel 89Zr-labeled DDS Device Utilizing Human IgG Variant (scFv): "Lactosome" Nanoparticle-Based Theranostics for PET Imaging and Targeted Therapy. Life (Basel). 2021, 11, 158; DOI:10.3390/life11020158.

We want to thank you for providing helpful comments, and we hope that the manuscript is now acceptable for publication.

Reviewer 4 Report

The anuscript entilted "Theranostics using indocyanine green lactosomes" reviews the possibility of using lactosomes as biocompatable nanoparticles in both treatment and imaging.

In my opinion this review is well structured and the literature in the field well explored.

In this sense it shoud be accepted after minor revisions.

Comments:

  • Abbreviations should be described when used for the first time
  • The authors should increase the qulaity of figures so that can be perceptible
  • Some figures have the font numeber too large. the authors should format according to the correct size

Author Response

We want to thank reviewer #4 for the valuable comments, which helped improve the quality of the revised manuscript. As shown in the responses, all comments were considered.

Comments from reviewer #4:

Comment 1:

Abbreviations should be described when used for the first time

Response 1:

Based on your suggestion, we have corrected the abbreviations.

Comment 2:

The authors should increase the qulaity of figures so that can be perceptible

Response 2:

We have revised the figures.

Comment 3:

Some figures have the font numeber too large. the authors should format according to the correct size

Response 3:

We corrected the font size accordingly.

We want to thank you for providing helpful comments, and we hope that the manuscript is now acceptable for publication.
